# Is COVID-19 a Real Incentive for Flu Vaccination? Let the Numbers Speak for Themselves

**DOI:** 10.3390/vaccines9030276

**Published:** 2021-03-18

**Authors:** Marcello Di Pumpo, Giuseppe Vetrugno, Domenico Pascucci, Elettra Carini, Viria Beccia, Anna Sguera, Maurizio Zega, Marcello Pani, Andrea Cambieri, Mario Cesare Nurchis, Floriana D’Ambrosio, Gianfranco Damiani, Patrizia Laurenti

**Affiliations:** 1Section of Hygiene, University Department of Life Sciences and Public Health, Università Cattolica del Sacro Cuore, 00168 Roma, Italy; giuseppe.vetrugno@policlinicogemelli.it (G.V.); domenico.pascucci@outlook.it (D.P.); elettra.carini@policlinicogemelli.it (E.C.); florianadambrosio@libero.it (F.D.); gianfranco.damiani@unicatt.it (G.D.); patrizia.laurenti@unicatt.it (P.L.); 2Fondazione Policlinico Universitario A. Gemelli IRCCS, 00168 Roma, Italy; anna.sguera@policlinicogemelli.it (A.S.); maurizio.zega@policlinicogemelli.it (M.Z.); marcello.pani@policlinicogemelli.it (M.P.); andrea.cambieri@policlinicogemelli.it (A.C.); 3Università Cattolica del Sacro Cuore, 00168 Roma, Italy; viria.beccia@gmail.com; 4Department of Woman and Child Health and Public Health-Public Health Area, Fondazione Policlinico Universitario A. Gemelli IRCCS, 00168 Roma, Italy; mariocesare.nurchis@unicatt.it

**Keywords:** flu vaccination, COVID-19, healthcare workers, adherence, vaccination coverage

## Abstract

Seasonal flu vaccination is one of the most important strategies for preventing influenza. The attitude towards flu vaccination in light of the COVID-19 pandemic has so far been studied in the literature mostly with the help of surveys and questionnaires. Whether a person chooses to be vaccinated or not during the COVID-19 pandemic, however, speaks louder than any declaration of intention. In our teaching hospital, we registered a statistically significant increase in flu vaccination coverage across all professional categories between the 2019/2020 and the 2020/2021 campaign (24.19% vs. 54.56%, *p* < 0.0001). A linear regression model, based on data from four previous campaigns, predicted for the 2020/2021 campaign a total flu vaccination coverage of 30.35%. A coverage of 54.46% was, instead, observed, with a statistically significant difference from the predicted value (*p* < 0.0001). The COVID-19 pandemic can, therefore, be considered as an incentive that significantly and dramatically increased adherence to flu vaccination among our healthcare workers.

Seasonal flu vaccination is one of the most important strategies for preventing influenza and reducing its healthcare, social, and economic impact [1,2,3,4,5]. Although influenza’s disease burden varies from year to year, evidence clearly shows that vaccination can reduce flu severity and prevent hospitalizations—critical considerations at a time when the health care system is burdened by coronavirus disease 2019 (COVID-19) [6].

COVID-19, therefore, should act as a great incentive for flu vaccination.

But is it really so? The attitude towards flu vaccination in light of the COVID-19 pandemic has so far been studied in literature mostly with the help of surveys and questionnaires [7,8,9,10]. These are very useful in identifying possible causes of hesitancy and in helping to plan vaccination strategies. A study by Wang et al., for instance, analysed COVID-19 vaccination intention in relation to flu vaccine uptake and classified the reasons for refusal as “suspicion on efficacy”, “effectiveness and safety”, “believing it unnecessary”, and “no time to take it” [7].

Whether a person chooses to be vaccinated or not during the COVID-19 pandemic, however, speaks louder than any declaration regarding his/her possible attitude towards it. A recent study analysed flu vaccine uptake in relation to COVID-19 vaccination intention and vaccine hesitancy among nurses [11].

We therefore proposed studying in our teaching hospital, Fondazione Policlinico Universitario “A. Gemelli” IRCCS (FPG), whether any significant increase in flu vaccination coverage occurred between last year’s flu vaccination campaign and this year’s campaign, marked by the COVID-19 pandemic.

As we can see from Table 1, there has been a statistically significant increase (tested with Pearson’s chi-square) in flu vaccination coverage across all categories, among both healthcare and non-healthcare workers. This could mean that COVID-19 acted as an incentive to flu vaccination beyond specific health-related education.

On a further note, younger generations tend to be more open to healthy lifestyles and good practices such as this [12].

Flu vaccination was offered to healthcare students across the two campaigns as well, with, respectively, 657 and 688 vaccinated students (+4.72%). The same vaccination time slots were offered across the two campaigns, and therefore a significant increase could not be observed, however much higher the demand was.

Many efforts have been made in the past to increase adherence to this public health practice among our healthcare workers, with steady but slow results up to 2019/2020, as shown in Figure 1.

Given the same conditions that had been present up to the start of the pandemic, a linear regression model, based on the data from the first 4 campaigns, predicted a total flu vaccination coverage of 30.35% (blue in Figure 1). A significant departure from the observed 54.46% coverage (orange in Figure 1) was found (*p* < 0.0001).

Although the analysis was performed without taking into account other possible confounders, the COVID-19 pandemic has been a major difference factor between 2020/2021 campaign and all other ones. It can, therefore, be regarded as an incentive that significantly and dramatically increased adherence to a good public health practice such as flu vaccination among our healthcare workers.

In conclusion, we hope that these results are indicative of a disposition towards future COVID-19 vaccination as well, as shown by other studies [11], even considering all the possible limitations connected to the analogy between this vaccine and the flu vaccine.

Let the numbers speak for themselves.

## Figures and Tables

**Figure 1 vaccines-09-00276-f001:**
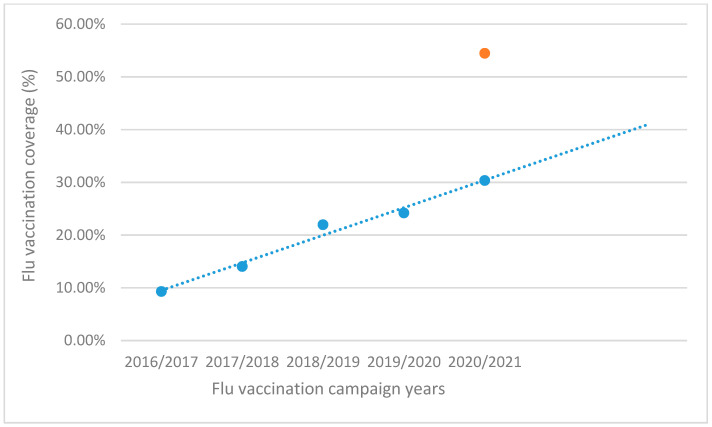
Flu vaccination coverage across 5 campaigns with a 4-campaign (2016/2017–2019/2020) linear regression line, 2020/2021 coverage observed (orange) vs. predicted (blue).

**Table 1 vaccines-09-00276-t001:** Vaccinated subjects by professional category campaign years and *p*-value (absolute and relative frequencies).

		2019/2020		2020/2021	
Professional Category	Vaccinated	Total	Vaccinated	Total	*p*-Value
Medical Doctors	483 (36.60%)	1320	819 (75.21%)	1089	<0.0001
Nursing staff	369 (17.35%)	2127	970 (48.04%)	2019	<0.0001
Other healthcare workers	222 (17.01%)	1305	881 (54.96%)	1603	<0.0001
Medical Residents	549 (45.22%)	1214	687 (55.90%)	1229	0.0025
Total healthcare workers	1026 (24.19%)	4241	2556 (54.46%)	4685	<0.0001
Administrative staff/non-healthcare workers	106 (10.01%)	1059	666 (54.06%)	1232	<0.0001

## Data Availability

Data presented in this study are available upon request from the corresponding author.

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
