# Peer review of "Is COVID-19 a Real Incentive for Flu Vaccination? Let the Numbers Speak for Themselves"

_vaccines, 2021, doi:10.3390/vaccines9030276_

Round 1
Reviewer 1 Report
In this manuscript, Marcello et al evaluates COVID-19 and a real incentive for flu vaccination. The current data does not satisfactorily addressed the conclusion made in the manuscript.
- The authors did not an intensive literature review on the selected topic.
- In this present situation is not new and comprehensive to the readers.
Author Response
Dear Reviewer,
We thank you for your suggestions.
here are our point-by-point responses.
In this manuscript, Marcello et al evaluates COVID-19 and a real incentive for flu vaccination. The current data does not satisfactorily addressed the conclusion made in the manuscript.
Thank you for your valuable suggestions. We are aware that there could be some confounders that might not have been taken into account given the nature of the research (i.e., Brief Report).
Anyhow, our analysis is a four-year based prediction where the fifth year is clearly distinguished from the others in terms of exposure to the COVID-19 pandemic. Moreover, the difference between what has been observed and predicted is ample both in terms of statistical significance and magnitude. We thank you again for the suggestion and we made sure to specify this in the text accordingly. The text has been, therefore, modified at the lines 85-93, 231-233 and 235.
The authors did not an intensive literature review on the selected topic.
Thank you for pointing out this issue. We conducted a deeper literature research in order to enrich the study, we added the references found and modified the text at the lines 85-89, 92-93, 237, accordingly.
In this present situation is not new and comprehensive to the readers.
Thank you for your comment. On the basis of the above-mentioned literature research, at the best of our knowledge, the considered topic seems to be groundbreaking and we only know of few recent similar experiences, which have been referenced in the text at the lines 85-89, 92-93, 237
Reviewer 2 Report
In this brief report, Di Pumpo and colleagues commented on the attitude towards flu vaccination in light of the COVID-19 pandemic. Based on the findings, flu vaccination coverage across all professional categories between the 2019/2020 and the 2020/2021 campaign has been significantly increased. By using a linear regression model and based on data from 4 previous campaigns, the authors predicted for the 2020/2021 campaign a total flu vaccination coverage of 30.35%. A coverage of 54.46% 27 was instead observed. These findings revealed that the COVID-19 pandemic could be considered as an incentive that significantly and dramatically increased adherence to flu vaccination among healthcare workers.
The report is well-written, except for some typos and grammatical errors that the author should correct before publication.
Author Response
Dear Reviewer,
We thank you for your suggestions.
here is our point-by-point response.
The report is well-written, except for some typos and grammatical errors that the author should correct before publication.
Thank you for your valuable suggestion. We corrected the typos and grammatical errors in the manuscript, accordingly.
Reviewer 3 Report
Good results presented from the teaching hospital. It would be great to see if these findings are generalisable. May be all hospitals have had dramatically increase adherence to flu vaccinations during the COVID-19 pandemic, at least among the healthcare workers.
Author Response
Dear Reviewer,
We thank you for your suggestions.
here are our point-by-point responses
Good results presented from the teaching hospital. It would be great to see if these findings are generalizable. May be all hospitals have had dramatically increase adherence to flu vaccinations during the COVID-19 pandemic, at least among the healthcare workers.
Thank you for your precious comment. We believe that our analysis is generalizable to similar contexts since the sample size is large, it is highly representative of all healthcare professional categories and it takes into consideration data deriving from five different consecutive years.
We can confirm that only few similar studies studying the adherence to flu vaccinations during the COVID-19 pandemic are present in literature at the moment. We reported a recent study that focuses on healthcare workers (i.e., nurses) as an example at the lines 92-93, 236-237
Round 2
Reviewer 1 Report
The revised manuscript improved significantly and the authors addressed all reviewers concern satisfactorily.